

# Evaluation of cognitive load in team sports: literature review

Joan Fuster[1], Toni Caparrós[1,2] and Lluis Capdevila[3]

[1] National Institute of Physical Education of Catalonia (INEFC), Universitat de Barcelona, Barcelona, Spain
[2] GRCE Research Group, National Institut of Physical Education of Catalonia (INEFC), Barcelona, Spain
[3] Sport Research Institute, Universitat Autònoma de Barcelona, Bellaterra, Barcelona, Spain

## ABSTRACT

In team sports, load management has become one of the most common areas of investigation, given that effective control over load is the key to being able to optimize performance and avoid injuries. Despite the constant evolution and innovation in the latest theories, we can see a clear tendency in load management that focuses on physiological and mechanical aspects and neglects its cognitive character, which generates the variability inherent in the performance of athletes in a changing environment. Indicators of response that inform methods of control over cognitive load can include cognitive, physiological and behavioral indicators. However, limited investigations exist to support the reliability of each indicator regarding cognitive load. For this reason, the objective of this literature review is to present strategies used to manage cognitive load in team sports, as well as the indicators utilized for such a proposition and their relationships in specific contexts.

## INTRODUCTION

Team sports have a dynamic and complex nature (*Lord et al., 2020*). The players perform intermittent, high-intensity activities such as repeated sprints, jumps, impacts, fights, changes of direction, accelerations and decelerations during training and competition (*Paulauskas et al., 2019*). In this regard, cognitive requirements are also considered crucial for optimal performance. These short, intense actions generally last less than 3 s, with moderate and long recovery periods (*Ben Abdelkrim et al., 2007*), and should be integrated into training strategies to promote adaptations that efficiently optimize athletic performance (*Schelling & Torres-Ronda, 2013*) through interaction with the training load.

Traditionally, planning for training and competition has centered on the management of physiological and mechanical aspects, defined as internal load (IL) and external load (EL), respectively. IL and EL are related, the first being defined as the individual psychophysiological response and the second as the external physical stimulus applied to the athlete during training or competition (*Soligard et al., 2016*). This process can be assessed by the control and management of certain indicators that could offer information about the load effect on the athletes (*Soligard et al., 2016*). In this context, however, the other elements that can affect the success of the athlete's training are neglected; for

Corresponding author
Joan Fuster,
joanfuscoll@gmail.com

example, the quantification of the cognitive effort (*Mujika et al., 2018*), defined as the volitional assignment of resources in order to respond to the demands imposed by a task. Depending on the degree of effort involved, cognitive resources can be exhausted which can cause mental fatigue, a psychobiological state provoked by prolonged periods of demanding cognitive activity (*Job & Dalziel, 2001*). Furthermore, *Van Cutsem, Marcora & De Pauw (2017)* and *Brown, Graham & Innes (2020)* conclude in their respective systematic reviews that cognitive effort causes a negative effect on physical performance.

Therefore, we consider that the cognitive load will be the relative amount of available mental resources that are invested in the resolution of a task. Cognitive load can be varied by changing the complexity of the assigned task but increasing its complexity will not involve a real increase of the individual's load if they do not employ enough effort to solve it (*Cárdenas, Conde-González & Perales, 2015*). For this reason, various authors describe how cognitive load is closely related to the emotional state of the athlete (*Camacho et al., 2020*; *Cárdenas et al., 2013*; *Cárdenas, Conde-González & Perales, 2015*; *Rottenberg & Gross, 2003*; *Vaughan, Laborde & McConville, 2018*). The regulation of these emotions involves processes through which individuals influence their emotions. These processes require effort, thereby draining cognitive resources (*Schmeichel, 2007*).

Team athletes perform in a high-entropy environment that causes them to constantly expend mental resources to respond to the demands of the tasks, which requires significant mental effort (*Cárdenas, Conde-González & Perales, 2015*). Furthermore, deliberate constant evaluation of the possible alternatives during a game can consume the resources of the system and promote fatigue, impairing performance (*Marcora, Staiano & Manning, 2009*). This cognitive demand combined to the effects of the internal and external load, can cause a maladaptation or rejection of the suggested load. This will directly influence whether the athlete is physically and/or mentally prepared for exposure to another training stimulus, known as the readiness to train/compete (*Gabbett et al., 2017*).

Cognitive load, together with its emotional dimension (*Alarcón et al., 2018*), requires structured planning and management that is complimentary with that of the physical load, providing information about how the planned sessions are being received. Other authors, such as *Camacho et al. (2020)*, *Cárdenas, Conde-González & Perales (2015)*, *Gabbett, Jenkins & Abernethy (2010)* and *García-Calvo et al. (2019)* agree with this reasoning. The main reason is that mismanagement of this load generates not only short-term effects, such as loss of performance (*Sansone et al., 2020*), reduction of technical abilities (*Gantois et al., 2019*) or loss of control over the real impact of the session (*McLaren et al., 2016a*); but also, long-term effects, such overtraining or burnout (*Goodger et al., 2007*).

Including this parameter in training plans will be key to balancing the readiness of the athletes, achieving optimal performance and preventing injuries. The team sports environment requires intense and constant cognitive activity (*Figueira et al., 2019*), employing various brain mechanisms to adapt to a changing environment (*Coutinho et al., 2017*). In fact, there is a need to include specific, suitable and objective information for successful planning (*Gabbett, Jenkins & Abernethy, 2010*), as well as the control of the

cognitive load of training (*Gonçalves et al., 2016*) and to define properly the variables involved on this process (*Soligard et al., 2016*). To this end, the current review aims to show, in a practical way, the strategies for the management of cognitive load in team sports, as well as the indicators used to monitor the load and the ways they are related in specific contexts.

## SURVEY METHODOLOGY

### Information sources

A search of the literature between 1970 and 2020 was performed using the following online databases: Medline (PubMed), SPORTDiscus, PsycINFO and Google Scholar system. The following keywords were used: team sports, cognitive load, mental load, psychological load, workload, mental fatigue, cognitive fatigue, psychological fatigue, together with Boolean operators such as "AND" and "OR". Furthermore, this literature review was performed in accordance with the Preferred Reporting Items for Systematic Reviews and Meta-Analyses (*Stewart et al., 2015*).

### Criteria of inclusion of the study

The titles and abstracts of all the articles were analyzed to determine the relevance of the publications for inclusion. Selection criteria for the articles were followed. The complete text of the publication was obtained to determine if it met the criteria for inclusion. In addition, the bibliographies of the selected articles were analyzed to find other relevant articles. Finally, for the current review, only those studies that are centered on the management of cognitive, mental or psychological load (in relation to the workload) and/ or the ones that analyzed cognitive, mental or psychological fatigue (the effect on work and performance) in the specific context of team sports, were included. Therefore, those articles that try to quantify the individual cost of mental resources, given some capabilities, while achieving a given level of performance in a task with specific demands were included.

### Criteria of exclusion of the study

Individual sports and studies in situations far from sports reality were excluded, as well as analyses in laboratory situations. Duplicate articles were eliminated and summaries, non-peer reviewed works, book chapters and opinion articles were also excluded. As a first step, the titles, abstracts and key words screening of the literature was carried out by the authors. In the second step, full-text articles of the relevant studies were screened while in the third step, the reference lists of the suitable articles and the review articles on the management of cognitive, mental or psychological load and/or fatigue were searched for additional articles. Any disagreement was discussed until consensus was reached.

## RESULTS

The initial literature search found 28,851 articles related to cognitive load, but only 1,947 were selected based on their title and abstract. After determining the content of the complete text, 1,926 articles were excluded for being unrelated or incompatible with the inclusion criteria, or both (Fig. 1). A total of 22 studies were included, which described the
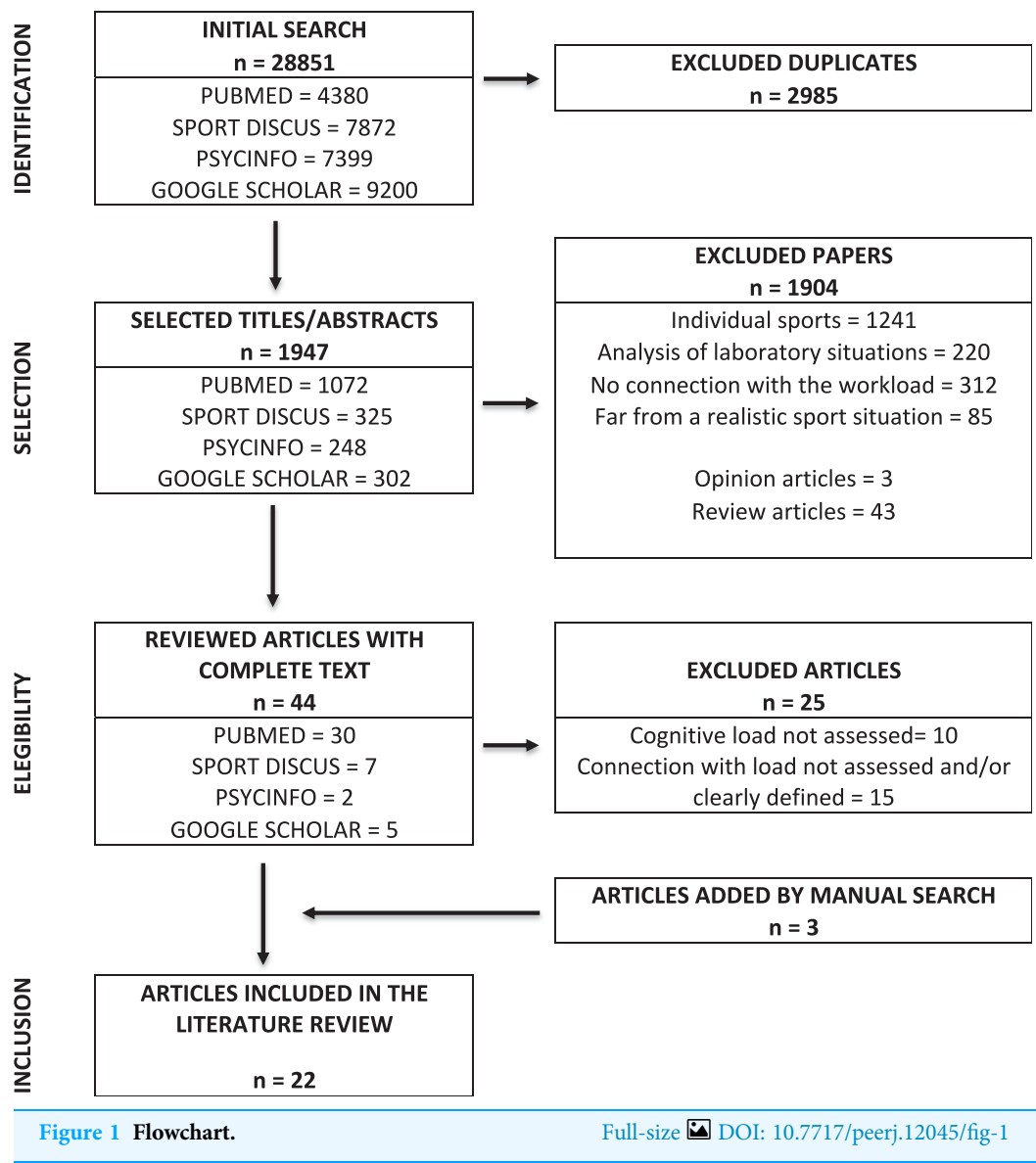

Figure 1 Flowchart.

management of cognitive load in different situations. These, following the classification of indicators described by *Capdevila (2001)*, were classified depending on the type of indicator of response analyzed, discriminating between cognitive (Table 1), a set of subjective indicators collected in a sporting situation, usually from written tests such as questionnaires, tests or self-reports; physiological (Table 2), indicators obtained with the aid of apparatus allowing the analysis of physiological or biochemical parameters; and behavioral (Table 3), indicators of the analysis of the observable behavior of the athlete, usually external motor behavior or verbal behavior.

## DISCUSSION

The authors use different methods to manage cognitive load, analyze specific and applicable information, exercise better control over training and optimize performance.
**Table 1 Managing of cognitive load with cognitive indicators.**

| Author | Year | n. | Objective | Task | Cognitive load variables | Internal load variables | External load variables | Measurement period | Results | Conclusion |
|---|---|---|---|---|---|---|---|---|---|---|
| Camacho et al. (2020) | 2020 | 22 university students with less than 2 years of experience playing basketball (20.81 ± 1.76) | To explore the impact of four task restrictions on mental load and their consequences for individual and team performance in small-sided basketball games (SSG). | 4 sessions of 3 × 3 SSG using the full court (5 × 2′, 2′ rec.) with restrictions: (A) Normal game (B) Temporal restriction (7″ to score) (C) Restriction of passes per possession (D) Conditions 2 and 3 simultaneously | Mental load (NASA-TLX, a.u.), affective response (PANAS questionnaire, a.u.) | Rating of perceived exertion (RPE) Borg scale 6–20 a. u., %HR peak, Edwards' heart rate zones | Ratio of successful offensive possessions (total num. of made shots/ total num. of offensive possessions), ratio of successful passes (successful passes/total num. of passes), ratio of rebounds (num. of captured rebounds/ num. of off and def missed shots) | During the task: %HR peak, Edwards HRZ, CE Variables After the task: NASA-TLX, PANAS, RPE | Statistical analysis showed that restrictions B, C, and D produced more mental load, provoking differences both in affective response and RPE, and consequently impairing performance. No significant differences were found in regards to the Internal Load of the sessions, extracted from the HR. | Restrictions imposed on SSG had a significant impact on subjective perceptions and cognitive and emotional responses. The analyzed restrictions are seen as an efficient way to increase the decision-making difficulty and, consequently, the mental load. It is valid to consider the collective performance, rather than the individual, as a way to generate a greater mental load |
| Kunrath et al. (2020) | 2020 | 18 amateur football players (21.8 ± 2.5) | To examine how mental fatigue (MF) has an influence on peripheral perception, tactical behavior and physical performance during a SSG in football | Two sessions were carried out: Control: Control visualization followed by SSG test task "Goalkeeper + 3 vs 3 + Goalkeeper" MF condition: Stroop task followed by SSG test task "Goalkeeper + 3 vs 3 + Goalkeeper" | Mental fatigue (MF) 100-mm VAS a.u. | Peripheral perception (°) | System of tactical assessment in football (FUT-SAT), tracking systems: Intensity zones in the distance covered, total distance covered, average V and max. V | Before and after the Stroop task: VAS MF, peripheral perception During the SSG: FUT-SAT and tracking system variables | MF diminished peripheral perception. Also, it induced a reduction in precision of tactical actions: offensive coverage, offensive unity, balance, concentration, mobility, and defensive depth and unity. Under MF conditions, players had elevated values for total distance, average V and time running at a moderate velocity. Greater high-intensity efforts were observed. | MF decreases peripheral perception provoking more errors for most tactical actions. The MF condition damages the cognitive and tactical aspects of the players' behavior, causing a compensatory increase in the physical performance. |

(Continued)

| Author | Year | n. | Objective | Task | Cognitive load variables | Internal load variables | External load variables | Measurement period | Results | Conclusion |
|---|---|---|---|---|---|---|---|---|---|---|
| *Sansone et al. (2020)* | 2020 | 12 semi-professional basketball players (21 ± 2 years) | To assess the technical-tactical, perceptive and mental demands of SSG in basketball. | 4 half-court 3x3 SSG sessions: (1) Long attack (3 × 4', 2' rec.) (2) Short attack (6×2', 1' rec.) (3) Long defense (3 × 4', 2' rec.) (4) Short defense (6 × 2', 1' rec.) | Mental fatigue (MF) 100-mm VAS a.u., Mental exertion (ME) 100-mm VAS a.u. | RPE with centiMax scale a.u., Perceived enjoyment 100-mm VAS a.u. | Quantitative technical-tactical video analysis | Right before the task: VAS MF During the task: T-T video analysis 5' after the task: RPE, VAS MF, VAS ME, VAS Enjoyment | Offensive tasks require greater ME in comparison with defensive ones. Short SSG induce better RPE and ME responses, as well as an increase in the demands of the game (possession, dribbling and shots). RPE is significantly associated with variations of mental fatigue provoked by the training exercises. | The short SSG induced a greater volume of offensive play and different tech-tac behaviors, (number of possessions and individual or collective actions). The offensive task is more demanding than the individual one, in terms of perceived effort and mental effort. |
| *García-Calvo et al. (2019)* | 2019 | 40 professional soccer players (17.32 ± 1.21 years) | To examine the effect of modifying the scoring system on the cognitive load perceived by football players due to training tasks | 4 drills (12'): (1) SSG, reduced game 5v5 (2) SSG 5vs5 with psycho-tactical load (scoring system) (3) Possession 5 vs 5 (4) Possession with psycho-tactical load (scoring system) | Mental load (adapted NASA -TLX, a.u.), 100-mm VAS mental fatigue a.u. | CR-10 RPE a.u., HR avg, HR max. | N° sprints, V avg (m/s) | During the task: HR, HR max, Sprints, V avg At the end of the task: NASA-TLX, VAS MF 30' later: RPE | The experimental task involves a significant increase of the load and the mental fatigue, in respect with the control task; having more impact on the SSG than in the possession drills. The physical load and the mental load behave in a different way: control situations, greater PL; experimental situations, greater ML. | The modified scoring system significantly affected the mental load and mental fatigue, with a stronger effect on the SSG than the possession drill. The physical and mental load behaved differently. |
| *Reina et al. (2019)* | 2019 | Adult amateur female basketball team (>18 years) | To quantify the training load *via* three types of instruments and establish possible relationships between the measurements utilized | Different training tasks were analyzed (a total of 120 tasks) | External subjective load test items (STATE): competitive load and cognitive involvement (scale from 0–5) a.u. | % Maximum Heart Rate | SIATE (Integrated System of Training Task Analysis) (Opposition degree, task density, field of play and percentage of performers), Player load | All averages were measured during training | The subjective load measurement maintains a statistically significant correlation with the variables of objective load and %MHR. The SIATE task is a reliable method of analysis to track the load of the players. | The use of the SIATE ecological system to register tasks gives us the same information as other more expensive information systems, such as inertial devices or HR bands. |

| Author | Year | n. | Objective | Task | Cognitive load variables | Internal load variables | External load variables | Measurement period | Results | Conclusion |
|---|---|---|---|---|---|---|---|---|---|---|
| *Alarcón et al. (2018)* | 2018 | 28 semiprofessional football players (20.07 ± 0.23 years) | To advance the study of the relationship between mental and physical load during athletic activities and see how this interaction influences athletic performance. | A football-specific task with acoustic signals that alternates sprints of 12 m, jogging displacements and passes. Objectives of passing precision. Counterbalanced design between two conditions: without mental load (AML) and with mental load (ML) 2×(3 × 3'–2' pause). | Mental load (adapted NASA -TLX, a.u.), Emotional state (Self-assessment manikin scale, SAM, a.u.) | Anxiety level (STAI questionnaire, a.u.) | Qualitative passing precision (observational) | During the task: Passing precision After repetition: NASA-TLX, SAM and STAI | The mental fatigue is significantly greater in the ML condition than in the ACM. This difference provokes a decrease of the precision. Both frustration and subjective anxiety predict passing performance. There is a tendency for the performance to improve over the course of the test. | The presence of the physical load simultaneous to the mental load had a negative effect on precision. The emotional states in ML condition predicted the performance in the motor task. |
| *Barrett et al. (2018)* | 2018 | 32 professional football players (25 ± 8 years) | To investigate the effects of playing position and contextual factors on the internal load experienced during games, measured *via* differential ratings of perceived exertion (dRPE) | The entire season was analyzed (38 games in total) | dRPE CR-100, a.u.: RPE-T (cognitive demand) | dRPE CR-100, a.u.: RPE, RPE-B (dyspnea), RPE-L (muscular effort of the lower body) | Playing position and level of opponents | 15'–30' post game: dRPE | The wings reported a substantially higher dRPE in comparison to the rest of the positions. Substantially higher RPE-T were registered in the games played against the best teams in comparison to the rest. | Significant differences in dRPE were observed between different playing positions, with the wings being the players who assume the greatest load. The RPE-T increases when the game is against the top teams in the league. |
| *Coutinho et al. (2018)* | 2018 | 10 amateur football players (13.7 ± 0.5 years) | To examine the effect of additional mental and muscular fatigue in physical and tactical behavior in football. | Three situations of 5 × 5 SSG, random order (3 × 6'–3' pause, for each condition): (1) Normal condition (2) Additional muscular fatigue (previous task of repeated direction changes, RCOD) (3) Additional mental fatigue (*Stroop color-word* task of 30' before the SSG) | Mental fatigue (MF) 100-mm VAS a.u. | CR-10 RPE a.u. | GPS Variables: Physical variables (Total distance, High ratio, Low ratio, Acc., Dec.) and position variables (Dyads distance, ApEn, Stretch Index, Longitudinal Sync and Lateral Sync) | During the task: GPS units Pre-post task: VAS MF, RPE | Mental fatigue increases the low ratio, the number of acc. and the longitudinal sync and provokes a decrease of the distance covered at high intensities. The physical profile during the task is more affected under mental fatigue conditions. Mental fatigue provokes a decrease of the longitudinal sync, stretch index and dyads distance. | The improvement of the levels of longitudinal synchronization after muscular fatigue, suggests the use of tactical tasks after intense exercises. Mental fatigue is the kind of fatigue that most decreases the physical performance of the task, so high intensity training is directly affected. |

(Continued)

| Author | Year | n. | Objective | Task | Cognitive load variables | Internal load variables | External load variables | Measurement period | Results | Conclusion |
|---|---|---|---|---|---|---|---|---|---|---|
| Alarcón, Ureña & Cárdenas (2017) | 2017 | 18 semiprofessional basketball players (21.35 ± 2.48 years) | To analyze the effects of mental fatigue on free throw efficiency in basketball players. | Three sets of 10 free throws were performed, with active recovery between them, and carrying out the following task: (a) Oddball control task (b) Experimental task: 2-back task (Cognitive Load medium–high) | Mental load (NASA-TLX adaptation, a. u.) | Degree of anxiety (STAI questionnaire, a.u.) | Free throw efficiency (num. of shots, scores/failed throws and % scores) | During the task: Free throw efficiency  At the end of the task: NASA-TLX and STAI | Statistically significant differences in the average NASA-TLX scores were found between the control group and experimental group. The participants in the experimental group exhibited worse performance than the participants in the control group. | There is a negative influence of mental fatigue on precision abilities, such as the basketball free throw. |
| Arruda et al. (2017) | 2017 | 12 professional basketball players (18.6 ± 0.5 years) | To examine the effect of playing games against different rivals on pre-game salivary testosterone concentration (T), precompetitive anxiety and pre- and post-game cortisol concentration (C). | Three games at home against three rivals of different competitive levels were analyzed (Difficult, 1st–3rd ranked, HM; Medium, 4th - 6th, MM; Easy, 7th–10th; EM) and a training situation (TS) | CSAI-2 cognitive anxiety, a.u. | Testosterone concentration (T), Cortisol concentration (C), CSAI-2 somatic anxiety a.u. | | Before the start of the game/ training: Saliva samples (T and C) and CSAI-2  After the game/ training: Saliva samples (T and C) and CSAI-2 | The concentration of T (pre) is significantly lower in the training situations than in the games. A significant change of C (pre and post) in all conditions was observed. A higher C (pre) in HM was also observed, compared to TS and EM, as well as a higher HM C (post) in comparison to EM, MM and training. The level of anxiety is lower in EM than in MM or HM. The cognitive anxiety increases exponentially, more than the somatic, depending on the rival. | The results suggest that playing against a high level opponent can cause a higher level of psychobiological stress, probably because the level of the opponent can be perceived as a threat to the social status of a determined hierarchy. |

| Author | Year | n. | Objective | Task | Cognitive load variables | Internal load variables | External load variables | Measurement period | Results | Conclusion |
|---|---|---|---|---|---|---|---|---|---|---|
| Coutinho et al. (2017) | 2017 | 12 amateur football players (15.9 ± 0.8 years) | To identify the effects of mental fatigue and the modification of additional reference lines in the physical and tactical performance of the players during SSG of football. | Four situations of 6 vs 6 SSG, with previous work (control task or motor coordination work to induce mental fatigue). Comparison of the performance in 4 conditions: 1) MF vs no MF, normal field (MEN) 2) MF vs MF, field with additional reference lines (#MEN) 3) no MF vs MF, normal field (CTR) 4) no MF, field vs no MF, field with additional reference lines (#CTR) | Mental fatigue (MF) 100-mm VAS a.u. | CR-10 RPE a.u. | CMJ (cm), positional and physical data with GPS (m, m/s) | Pre task cont./ exp.: VAS MF, RPE and CMJ<br><br>Post task cont./exp. (pre SSG): VAS MF, RPE and CMJ<br><br>During SSG: GPS<br><br>Post SSG: VAS MF and RPE | After the experimental task, higher values were found in MF, VAS and RPE than in the control condition. MEN showed an increase of RPE in comparison to CTR. The CTR condition provokes an improvement if GPS positional variables in comparison to MEN. Adding the reference lines to the field provokes a deterioration of the physical GPS variables and an increase in RPE. Furthermore, adding them in MF situations provokes a deterioration of the positional GPS variables. | The results showed that MF affects both the capacity to use environmental information and the position of the players. The additional reference lines could have improved the use of less relevant information to guide actions during the #MEN condition. The motor coordination task induces a similar MF to the levels reported in previous studies using Stroop tasks. |
| Vallés, Fernández-Ozcorta & Suero (2017) | 2017 | 12 professional basketball players (21.91 ± 4.81 years) | To analyze the technical-tactical training load assessment (VSC) and its connection to the parameters of internal load. | All trainings during the last 2 months of competition were analyzed (Women's league). | VSC test items: competitive load and cognitive implication (scale 0–5) a.u. | CR-10 RPE a.u. and TQR | VSC test items: Degree of obstacle, task density, num. of participants and game field. | During training: subjective assessment of the load 20′ post training: RPE The day after training: TQR | Except for the degree of obstacle of the task, the VSC scale shows a high correlation with the RPE. There are no correlations between the VSC scale and the TQR. The task density explains 91,4% of the RPE. | The results revealed that the VSC scale shows a high correlation with the RPE, but not with the TQR. This indicates that the VSC can be a good tool to calculate the internal load in basketball. |

(Continued)

| Author | Year | n. | Objective | Task | Cognitive load variables | Internal load variables | External load variables | Measurement period | Results | Conclusion |
|---|---|---|---|---|---|---|---|---|---|---|
| Veness et al. (2017) | 2017 | 10 professional cricket players (21 ± 8 years) | To investigate the acute effects of a mentally exhausting task on cricket-relevant physical and technical tasks. | Three specific physical tests were carried out in three different sessions: Cricket run-two test, Batak lite reaction time test and Yo-Yo-IR1 test; preceded for 30' by a control task or a Stroop task (induces MF) | Mental fatigue (MF) 100-mm VAS a.u. | CR-10 RPE a.u., Motivation 100-mm VAS a.u. | Results obtained in the specific technical and physical tests | Before the task: VAS MF, VAS motivation After the task: RPE, VAS MF | Significant differences in the perception of mental fatigue were observed after the treatment. The specific physical Yo-Yo-IR1 and Cricket run-two tests, as well as the RPE, were affected by MF. In the case of the Batak lite test, the change was moderated, but their performance was not affected. | The mental fatigue induced by a Stroop task significantly affects cricket-relevant technical and physical performance. |
| Badin et al. (2016) | 2016 | 20 professional football players (17.8 ± 1 years) | To assess the effects of mental fatigue on the technical and physical performance in SSG of football. | Two sessions of 5 vs 5 SSG without goalkeepers (2 × 7', 1' rec), with a preceding exposure to 30' of control or Stroop task (induces MF). | Mental fatigue (MF) 100-mm VAS a.u., Mental Effort (ME) 100-mm VAS a.u. | %HR max, CR-10 RPE a.u., Motivation (M) 100-mm VAS a.u., Physical Fatigue (PF) 100-mm VAS a.u. | GPS Variables: Velocity zones, total distance, repeated sprints and accelerations. % passing precision (successful/ total), % effective challenges (successful/ total), % possessions (positive/ negative) and % participations (positive/ negative) | Before the task: VAS MF, VAS PF, VAS M, VAS PF After the task: VAS MF, VAS ME, VAS PF, VAS M, RPE During SSG: %HR max, GPS variables After the SSG: VAS MF, VAS PF, VAS ME, RPE | The MF and ME increased significantly after carrying out the Stroop task, while motivation was similar in both conditions. The RPE was greater in the condition of mental fatigue, although the HR was greater in the control situation. The mental fatigue does not have a very clear effect on most of the variables of physical performance, but it damages the technical performance variables. | Mental fatigue did not affect physical performance, despite the increase in perceived effort. In contrast, mental fatigue impaired offensive and defensive technical performance. |

| Author | Year | n. | Objective | Task | Cognitive load variables | Internal load variables | External load variables | Measurement period | Results | Conclusion |
|---|---|---|---|---|---|---|---|---|---|---|
| McLaren et al. (2016) | 2016 | 29 professional rugby players (24 ± 3 years) | To investigate the application of dRPE in the training for team sports. | The trainings of an intensive 6-week period were monitored. A linear increase of the load was produced in the first 3 weeks and after that it was gradually reduced. | sRPE CR-100, a. u.: RPE-T (cognitive demands) | sRPE CR-100, a. u.: RPE, RPE-B (dyspnea), RPE-L (muscular effort in the lower body), RPE-U (muscular effort in the upper body) | Classification of the training typology: HIIT (High-intensity intervals training, RHIE (Repeated high intensity efforts), SkCond (Conditioning based on skills), Skills, RT (resistance task) and URT (Upper body resistance physical task) | 15' post session: sdRPE | Differences in the sRPE between sessions are observed depending on the training typology. The dRPE loads combined to explain the 66-91% of variance of the sRPE loads. The strongest associations are found between: sRPE-L for HIIT, sRPE-U for RHIE, sRPE-U for RT and sRPE-T for Velocity and Skills. | The dRPE can provide a detailed quantification of the internal load during training activities. The knowledge of the connections between sRPE and dRPE can isolate the specific perceptive demands of the different training methods. |
| Smith et al. (2016) (Study 2) | 2016 | 14 amateur football players (19.6 ± 3.5 years) | To investigate the effects of mental fatigue on technical performance evaluated via football-specific tests | Two sessions of football-specific technical tests: LSPT (Loughborough specific pass test) and LSST (Loughborough specific shot test). These were preceded by two 30' situations, a control and a Stroop task (induces mental fatigue) | Mental fatigue (MF) 100-mm VAS a.u., Mental Effort (ME) 100-mm VAS a.u. | Motivation (M) 100-mm VAS a.u. | Results obtained in the football-specific technical tests | Before the task: VAS MF, VAS M, VAS ME After the task: VAS MF, VAS ME, Test results | MF and ME increased significantly after carrying out the Stroop task, although motivation was similar in both conditions. The performance time in LSPT did not vary between conditions, but the penalization time (errors) increased significantly in the condition of MF. MF also provokes a decrease of the velocity of throw and precision in LSST. | Mental fatigue damages football-specific technical performance. |

(Continued)

| Author | Year | n. | Objective | Task | Cognitive load variables | Internal load variables | External load variables | Measurement period | Results | Conclusion |
|---|---|---|---|---|---|---|---|---|---|---|
| Gabbett, Jenkins & Abernethy (2010) | 2010 | 16 professional rugby players (17.3 ± 0.9 years) | To investigate the physiological, cognitive and skill demands of "on-side" and "off-side" games in elite rugby players | Two 8′ 8 *vs* 8 SSG sessions, in random order. - "off-side" SSG (passing the ball to a player that is offside is allowed) - "on-side" SSG (the ball can only be passed to the players that are onside) | Cognitive RPE a. u. | Heart Rate and RPE-10 a.u. | GPS variables: total distance, relative distance, accelerations (in three levels), velocity (in five levels) and recovery (time, in three levels) Technical variables (well or poorly executed): passes, receptions, disposal efficiency and involvements (objective of the game) | During the task: GPS variables, technical variables and HR When the task ends: RPE and Cog.RPE | "Off-side" games have a greater amount of involvements, passes and effective passes. Moreover, they also provoke a greater total covered distance, and increase minor and moderate accelerations as well as the distance covered at low, medium and high velocities. Significant differences in disposal efficiency were not found. In "off-side" games recovery is lower. Cog. RPE is lower for "on-side" games. | The results show that "off-side" games provide a greater physiological and skills demand than "on-side" games. For this reason, they are a good tool to develop physical conditioning and technical skills. Even so, "on-side" games show a greater cognitive load. |
| Farrow, Pyne & Gabbett (2008) | 2008 | 13 amateur Australian football players (16.7 ± 0.5 years) | To compare the physiological, cognitive and skill components in open and closed exercises, within game-based drills used in Australian football. | Two training sessions in two days. The training tasks used were 3-man weave, square kicking drill and diamond kicking drill. These use open exercises in one session and closed exercises in the other one. | RPE Cognitive a. u. | Heart rate, blood lactate and RPE-10 a. u. | GPS variables and technical variables | During the task: GPS variables, technical variables and HR At the end of the task: RPE, lactate and Cog. RPE | Two of the three open exercises were more demanding in terms of the total covered distance, RPE and relative intensity. All the open exercises had more efforts of moderate velocity. Differences in the concentration of lactate were not found. The HR was greater in the open format of the third exercise. The open exercises were more technically demanding. Higher cognitive load scores were obtained in the open exercises. | The open exercises were generally more cognitively and physically demanding than the closed ones, commonly used in Australian football. Open exercises should be prescribed to obtain greater cognitive and physical training loads in a game-specific context. |

| Author | Year | n. | Objective | Task | Cognitive load variables | Internal load variables | External load variables | Measurement period | Results | Conclusion |
|---|---|---|---|---|---|---|---|---|---|---|
| Mashiko et al. (2004) | 2004 | 37 university rugby players (20.3 ± 1.5 years) | To know the relationship between physical and mental fatigue in rugby players after a game, based on their playing position | Every game of the competitive season was analyzed | POMS: Profile of mood states questionnaire (to evaluate mental fatigue) a.u. | Biochemical parameters in the blood and opsonic activity in the serum | | All data was collected twice: in the morning on the day of the match (fasting condition) and right after the match | Differences were not observed in the changes in the biochemical parameters, except for ureic nitrogen in blood, between forwards and 3/4. In regards with the correlation between mental and physical fatigue, in the forwards, the changes in the POMS scores show a positive correlation with the levels of enzymes of musculoskeletal origin. In the 3/4 the changes in the POMS scores show a positive correlation with the changes in the levels of the parameters related to the lipids. | The exercise load differs depending on the position played during the rugby game. It can cause important differences in the connection between mental and physical fatigue. |

**Table 2 Managing of cognitive load with physiological indicators.**

| Author | Year | n. | Objective | Task | Cognitive load variables | Internal load variables | External load variables | Measurement period | Results | Conclusion |
|---|---|---|---|---|---|---|---|---|---|---|
| *Gantois et al. (2019)* | 2019 | 20 professional football players (22.6 ± 3.3 years) | To analyze the effect of mental fatigue on decision-making in football players | Three different tests were carried out over 3 weeks (1 week between sessions): "Stroop task", followed by a warm-up and a SSG. Stroop tasks: control (CON), 15′ ST (15ST) and 30′ ST (30ST) | Heart rate variability, HRV (RMSSD, SDNN and pNN50) and reaction time, RT (s) | Urine osmolality (Armstrong's scale), TQR questionnaire (total quality of recovery, a.u.), CR-10 sRPE a.u. | Index of Passing Decision-making (PDM) (%) | Pre/Post Stroop Task: Urine, TQR, HRV, RPE<br><br>During SSG: PDM, TR 30′ post SSG: sRPE | There are no significant differences between HRV, TQR, Urine, sRPE and the different Stoop tasks. All participants showed a decrease of PDM in conditions of 30ST in comparison to 15ST and CON. The RT increases significantly with mental fatigue. | The mental fatigue induced by the Stroop task damages the decision making of the football players and therefore their performance. |

To do so, they try to quantify, through indicators, the individual cost of mental resources, given certain capacities, while reaching a certain level of performance in a task with specific demands. Cognitive load management methods can be used before, during or after task performance. These cognitive load management methods are based on the analysis of these response indicators, usually classified as cognitive, physiological and behavioral indicators (*Capdevila, 2001*), so this review will address all three types.

## Cognitive indicators

### NASA-TLX

The NASA-TLX questionnaire has proven sensitive to the mental load in a variety of cognitively demanding tasks, such as piloting an aircraft or laboratory tasks (*Luque-Casado et al., 2016*). This scale analyses the subjective perceived work according to six dimensions: mental demand (perceptive and mental effort), physical demand (degree of physical activity), temporal demand (perceived pressure related to decision-making speed), effort (the combination of the mental and physical effort needed) and frustration (the negative emotions perceived). This offers an overall score of the workload (from 0 to 100 points, a.u.) based on the average of the six dimensions.

In all studies analysed, the NASA-TLX questionnaire was significantly sensitive to changes in the cognitive load in different team sports situations. In tasks that involve mental fatigue, significant differences are shown between the control and experimental groups (*Alarcón, Ureña & Cárdenas, 2017*; *Alarcón et al., 2018*). In the study by *Camacho et al. (2020)* performance on training tasks under temporal and quantity-of-technical-movement restrictions was analysed and the most specific management tool for the increased cognitive load was the NASA-TLX. Likewise, *García-Calvo et al. (2019)* also used this tool to identify the increase of the cognitive load as a function of the modification of the scoring system.

Measurement Period: After the session/competition.

**Table 3 Managing of cognitive load with behavioral indicators.**

| Author | Year | n. | Objective | Task | Cognitive load variables | Internal load variables | External load variables | Measurement period | Results | Conclusion |
|---|---|---|---|---|---|---|---|---|---|---|
| *Gantois et al. (2019)* | 2019 | 20 professional football players (22.6 ± 3.3 years) | To analyze the effect of mental fatigue on decision-making in football players | Three different tests were carried out over 3 weeks (1 week between sessions): "Stroop task", followed by a warm-up and a SSG. Stroop tasks: control (CON), 15′ ST (15ST) and 30′ ST (30ST) | Heart Rate Variability, HRV (RMSSD, SDNN and pNN50) and reaction time, RT (seconds) | Urine osmolality (Armstrong's scale), TQR questionnaire (Total Quality of Recovery, a.u.), CR-10 sRPE a.u. | Index of Passing Decision-making (PDM) (%) | Pre/Post Stroop Task: Urine, TQR, HRV, RPE During SSG: PDM, TR 30′ post SSG: sRPE | There are no significant differences between HRV, TQR, Urine, sRPE and the different Stroop tasks. All participants showed a decrease of PDM in conditions of 30ST in comparison to 15ST and CON. The RT increases significantly with mental fatigue. | The mental fatigue induced by the Stroop task damages the decision making of the football players and therefore their performance. |
| *Moreira et al. (2018)* | 2018 | 32 high level basketball players (15.2 ± 1.2 years) | To examine the effect of mental effort on physiological changes and technical performance in SSG. | Two 4 × 4 SSG trainings, separated by a week (4 × 2′30–1′ pause). They are preceded by two situations: (1) Control: 30′ control treatment (2) Experimental: 30′ Stroop Task | Reaction time, RT (s) | RPE session (CR-10, a.u.), HR, saliva samples: concentration of testosterone (T), cortisol (C) and alpha-amylase (sAA) | Performance parameters through video analysis. | Pre task cont./exp.: Saliva During the task cont./exp.: RT Post task cont./exp. (pre SSG): Saliva During SSG: Performance parameters and HR Post SSG: Saliva (15′ post) and RPE (30′ post) | The RT decreased significantly during the test, except during the last 5′. There's a direct correlation between the number of turnovers and the experimental task. Thera are an increase of the concentrations of T and sAA from the pre-control treatment to the post SSG. There is little difference between the treatments regarding the sRPE and the HR. | Mental fatigue induced by a Stroop task affects the performance of the task negatively. This also provokes a modulation of the endocrine response (it alleviates the concentrations of T and sAA, in comparison to the control condition). Between sRPE and HR we do not see very clear differences. |
| *Scanlan et al. (2013)* | 2013 | 12 basketball players (25.9 ± 6.7 years) | To determine the influence of physical and cognitive factors in the development of reactive agility in basketball players. | Three basketball-specific tests were carried out: multiple sprint test, Change of Direction Speed Test and Reactive Agility Test | Reaction time and decision-making time (s) | | 5, 10, and 20 m sprint times (s), max. V (m/s), reactive agility time (s), closed-skill agility time (s) | All measures were carried out during the basketball-specific tests | The simple and stepwise regression analyses determined the individual influence of each predictive variable and the best predictive model for reactive agility time. Through the stepwise model, the reaction time is identified as the only predictive variable. | The cognitive measures had the greatest influence on the reactive agility performance of the basketball players |

## 100 mm VAS (MF and ME)

The VAS scale (Visual Analogue Scale) offers specific information about a characteristic or attitude which is identified along a continuum of values and cannot easily be directly measured (*Gould et al., 2001*). It has a unidimensional format, charted in a 100 mm straight line, with limits identified as the perceived minimum (0) and maximum (100). The subjects only need to make a mark on the line that indicates their relative perception of their current situation. The distance will be measured in millimeters (from left to right) and this will be the subjective reference value.

This tool has been used frequently in the bibliography to describe the fatigue and lack of energy caused by a cognitively demanding activity (mental fatigue, MF) and/or the degree of effort employed to perform a cognitively demanding activity (mental effort, ME). It has been proved that the scale is a valid, reliable method to measure both MF and ME (*Lee, Hicks & Nino-Murcia, 1991*).

The first investigations demonstrated that the increase of perceived indicator VAS MF damaged specific technical performance (*Smith et al., 2016*; *Badin et al., 2016*). In Badin's study (2016), effects of VAS MF on physical abilities were not found, in contrast to subsequent studies which related VAS MF to the loss of both technical and physical abilities (*Veness et al., 2017*; *Coutinho et al., 2017*; *Coutinho et al., 2017*). *Sansone et al. (2020)* adds that within this loss of abilities, offensive tasks cause more significant VAS MF and ME than defensive tasks. Differing a bit from what has been previously described, *Kunrath et al. (2020)* shows that VAS MF reduces technical, tactical and cognitive abilities, provoking a compensatory improvement of the performance.

Measurement Period: After the session/competition.

## CSAI-2

CSAI-2 (Competitive State Anxiety Inventory 2) is a test that aims to estimate the cognitive and somatic anxiety levels of the players, as well as their levels of self-confidence. Cognitive anxiety refers to the negative feelings that the subject has about his or her performance and the consequences of the outcome. Somatic anxiety, on the other hand, refers to the perception of physiological indicators of anxiety such as muscle tension, increased heart rate, sweating and stomach discomfort. It has a questionnaire format consisting of 27 items, with nine items for each subscale. Each of them is classified on a four-point Likert-type scale, which yields scores from nine to 36 in each subscale. A higher score related to cognitive and somatic anxiety indicates a higher level of anxiety (*Martens et al., 1990*).

*Arruda et al. (2017)* study suggests that cognitive anxiety increases as a function of the level of the opponent, causing more psychological stress. Furthermore, it highlights that cognitive anxiety increases exponentially depending on the rival, more than somatic anxiety. This information could offer an approximation to the concept of cognitive load, dissociating the somatic state from the cognitive one.

Measurement Period: After the session/competition.

## POMS

The POMS (Profile of Mood States) questionnaire was proposed by *McNair, Losr & Droppleman (1971)* with the aim of evaluating the mental fatigue related to physical effort. The questionnaire is classified on a Likert scale and it contains 65 items that provide measures of six specific mood states: tension, depression, anger, vigor, fatigue and confusion. These factors can be combined to create a compound measurement of mood state by adding the five negative factors and subtracting the positive factor of vigor. Further, a 100-point baseline score is added to prevent negative scores (*Raglin, Morgan & O'Connor, 1991*).

It has been proved that POMS is a reliable and valid questionnaire to measure affective features, mood state and emotions (*Lin, Hsiao & Wang, 2014*). That is why *Mashiko et al. (2004)* use it as a test of mood state in order to evaluate mental fatigue. In their study they could observe significant changes between different game positions in rugby games, obtaining important differences between the relation of mental and physical fatigue.

Measurement Period: After the session/competition.

## Cognitive RPE

The RPE (rating of perceived exertion) is a valid method for quantification of the effort expended in an athletic training activity during a wide variety of exercise types (*Foster et al., 2001*). Depending on the question the athlete is asked, distinctive subjective variables can be measured. These differential ratings of perceived exertion (dRPE) can provide additional information to that obtained by a single measurement (*Gil-Rey, Lezaun & Los Arcos, 2015*). In this review, we focus on cognitive RPE (RPE-T in the classification of dRPE), which is the one that answers the question "How much mental effort and decision-making has this task required?" To quantify it, two scales are used, CR-100, also known as "centiMax", (*Borg & Borg, 2001*), scored from 0 to 100 and CR-10 (*Borg, Hassmen. & Langerstorm, 1985*), varying from no exertion (0) to maximum effort, such as a competition (10). The important difference between them is that the CR-10 uses whole numbers corresponding to verbal anchors and the CR-100 does not (*Fanchini et al., 2016*).

In *Farrow, Pyne & Gabbett (2008)* study it was shown that cognitive RPE (CR-10) was sensitive to tasks that involved increased decision-making in game-like situations. Having followed this line of research and found results that agree with the mentioned study, *Gabbett, Jenkins & Abernethy (2010)* justify the use of cognitive RPE as valid for control of cognitive load, but since it is a subjective measurement it would require more objective methods to be compared and measured. In order to relate the subjective data obtained to some objective data, *McLaren et al. (2016b)* finds strong associations between RPE-T multiplied by the session time (sRPE-T) and the tasks focused on speed and specific abilities, the latter being open tasks focused on the game. This RPE-T increases significantly when the game is against the top teams of the league (*Barrett et al., 2018*). Therefore, we can say that according to the results of these four studies, cognitive RPE

is sensitive to open tasks that seek to replicate the competitive game, creating an uncertain environment.

Measurement Period: After the session (recommended 15–30′ later).

### SIATE/VSC

SIATE (*Sistema Integral de Analisis de Tareas de Entrenamiento*, from its Spanish initials) and VSC (*Valoración Subjetiva de la Carga*, from its Spanish initials) are ecologic systems for quantifying the load of a basketball training session (*Vallés, Fernández-Ozcorta & Suero, 2017*; *Reina et al., 2019*). They are intended to provide a holistic view of training, taking into account more than just internal and external load.

*Ibáñez, Feu & Cañadas (2016)*, in order to quantify training using direct observations, designed a methodological system to register and subsequently analyze different factors that are relevant to the athletic process. For this purpose, they created an observational survey known as Integrated System of Training Task Analysis (SIATE). This survey records six variables: degree of opposition, density of task, percentage of simultaneous participants, competitive load, space of play and cognitive involvement. The variables are scored from one (minimum load) to five (maximum load). The sum of the scores provides a measurement of the total load of the task.

*Coque (2009)* suggests a tool that, similarly to the previous one, aims to evaluate the training load using a direct observation. The analysis system is the Subjective Evaluation of the technical-tactical training load (VSC), in which six variables are recorded: degree of obstacle, density of the task, percentage of simultaneous executions, competitive load, field of play and cognitive involvement. The variables are scored from one (minimum load) to four (maximum load). The sum of the scores provides a measurement of the total load of the task.

Both of them can be used to calculate the total load of the task, which would be the total of the assessed values, and the load weighted for the length of the task, which would be calculated by multiplying the total load by the useful time of the task in minutes. The latter shows more precision in the real load of the task. *Reina et al. (2019)* shows that the SIATE organic system gives us the same information as that recorded by inertial devices or HR monitors. Also, *Vallés, Fernández-Ozcorta & Suero (2017)* show that VSC has a strong correlation with RPE. Studies that analyze correlations between individual variables to see the percentage of the variability as a function of the cognitive load have not been found.

Measurement Period: After the session/competition.

## PHYSIOLOGICAL INDICATORS

### Heart rate variability (HRV)

Heart Rate Variability is defined as the temporal variation of the heart rate during a specified time period (*Capdevila & Niñerola, 2006*). In the simplest way, HRV has been analysed within the time domain, but more complex evaluations include an analysis within the frequency domain and nonlinear methods (*García Manso, 2013*).

The Autonomic Nervous System (ANS) is responsible for the regulation of the HRV through parasympathetic and sympathetic modulation (*Bricout, DeChenaud & Favre, 2010*), the balance of which is disrupted after changes in the training load (*Pichot, Busso & Roche, 2002*). This is the reason why many studies define HRV analysis as a useful, non-invasive method to evaluate the function of the ANS (*Bellenger et al., 2016*; *Bosquet, Merkari & Arvisais, 2008*; *Hynynen et al., 2006*; *Parrado et al., 2010*). The findings in the *Thayer et al. (2009)* study suggest an important relationship between cognitive performance and HRV, reaffirming the relevance of this information to measure the effect of cognitive load.

Even though the literature in control situations suggests that HRV is very sensitive to demands on the cognitive load (*Luque-Casado et al., 2016*), no significant differences have been found in HRV comparing HRV control situations and mental overload (*Gantois et al., 2019*).

Measurement period: Before the session.

## BEHAVIORAL INDICATORS

### Reaction time or response time

Reaction time is calculated as the time from the beginning of the stimulus until the corresponding response of the participant (*Gabbett, Kelly & Sheppard, 2009*). This indicator depends mainly on, but may be affected by age, gender or duration of the stimulus (*Der & Deary, 2006*), on cognitive processes, which means that mental fatigue could be inhibitory (*Huijgen et al., 2015*) and could increase the response time. It should be noted, however, that beyond a certain duration and/or intensity of exercise, muscle fatigue induces an increase in reaction time which may be due to a decrease in cognitive performance. fatigue induces an increase in reaction time that may be due to a decrease in cognitive performance (*Chmura, Nazar & Kaciuba-Uscilko, 1994*).

In *Scanlan et al. (2013)* response time is identified as the only variable that predicts the time of reactive agility in the phased model. This serves us to determine the influence of physical and cognitive factors on the development of reactive agility in basketball players. *Gantois et al. (2019)* found a significant increase in reaction time in relation to increased mental fatigue induced by the Stroop task. This increased reaction time impairs the decision making of soccer players and thus their performance. Coinciding with these results, *Moreira et al. (2018)* saw an increase in reaction time caused by mental fatigue in a Stroop task. Reaction time underwent a significant decrease during the test, except for the last 5 min in which it was maintained

Measurement period: During a task previous to training/competition.

### Decision-making time

Making a successful decision depends on the ability of the player to identify, select and then execute the correct action in response to the postural signs of opponents or teammates, recognizing significant patterns in the game and determining the situational probabilities (*Roca et al., 2013*; *Williams et al., 2011*). Decision-making time is determined as the time interval between the first identifiable contact of the stimulus-player and the

first identifiable contact that initiates the participant's response (*Gabbett, Kelly & Sheppard, 2009*). This cognitive ability to make fast and precise decisions is fundamental for success in football (*Smith et al., 2016*), an observation that can be applied to the rest of the team sports.

In *Scanlan et al. (2013)* decision-making time is identified as a variable which is highly associated with agility time. This cognitive measurement, together with reaction time, was the one that influenced the basketball players the most. *Gantois et al. (2019)* showed that induced mental fatigue damaged the decision-making process, provoking a decrease in performance. These findings agree with those of *Smith et al. (2016)*, determining that mental fatigue affected the decision-making precision and time of the football players.

Measurement period: During the task.

## CONCLUSIONS

### Relation with internal and external load

Cognitive factors interact with physiological and mechanical factors which are present during training and competition. For this reason, distinguishing their connections can be a significant element in the research of load management in sport (*Soligard et al., 2016*). In this context, the loads that the athletes assume involve stress and provoke changes in their physical and psychological well-being. Furthermore, understanding the interaction between these loads, perceptive well-being and readiness for training or competition will provide us with significant individual training prescriptions (*Gabbett et al., 2017*).

The IL measured through RPE is significantly connected to variations in cognitive load imposed by the training exercises, as evaluated by cognitive indicators such as the NASA-TLX questionnaire (*Camacho et al., 2020*) or the VAS MF scale (*Badin et al., 2016*; *Sansone et al., 2020*; *Veness et al., 2017*). This increase in cognitive fatigue has a direct impact on the internal load, provoking a modulation of the endocrine response attenuating the concentration of testosterone and alpha-amylase, markers of the activity of the mesolimbic pathways and the sympathetic nervous system, respectively (*Moreira et al., 2018*). There were no studies showing significant correlations between the internal load taken from the HR and the cognitive load taken from the NASA-TLX questionnaire (*Badin et al., 2016*), except for the study conducted by *Farrow, Pyne & Gabbett (2008)* which showed increases in the RPE-Cog and the HR during open exercises.

Moreover, the EL recorded through tracking systems shows that players under high cognitive load conditions (VAS MF) register higher values for total distance, average speed and run time at a moderate speed, demonstrating higher intensity efforts (*Coutinho et al., 2017*; *Kunrath et al., 2020*). This condition of mental fatigue damages aspects of the tactical behavior of the players, causing a compensatory increase in the physical performance (*Kunrath et al., 2020*). Position variables, taken from GPS analysis, are also affected in high cognitive load conditions, provoking a decrease of the longitudinal synchronization (coordination of longitudinal movements of the players during the game), of the stretch index (represented by the average distances of each player from the geometric gravity centers of the team) and of the distance between dyads (distance between

a pair of players that share the same environment and intend to reach the same team objective) (*Coutinho et al., 2018*). This deterioration of physical performance (physical variables) and elapsed synchronization time (position variables) should alert us that cognitive load should be considered a variable that can be controlled with the objective of improving collective behavior.

The models of training load control that take into account cognitive load such as the CSE or VSC can offer the same information as more difficult methods (*Reina et al., 2019*; *Vallés, Fernández-Ozcorta & Suero, 2017*). However, we must take into account that they are both subjective recording systems.

## Practical applications

To optimize performance and exert more control over training processes and/or competition in team sports, it is necessary to have load control strategies which include cognitive load in the monitoring cycles of the athlete. However, there is limited literature related to this concept.

NASA-TLX, VAS MF or Cognitive-RPE analysis strategies are considered valid to measure the impact of the cognitive load in team sports. Other control methods like VFC, reaction time or decision-making time need scientific evidence. In this sense, there is a clear lack of studies which use objective tools to measure the cognitive load.

The evaluation of external, internal and cognitive demands in an isolated way would be potentially problematic, because we would only obtain information on the provided stimulus, the processed stimulus or the players responses without examining the inherent connections. Even though it is accepted that external, internal and cognitive demands are separate constructs, they must be analyzed and interpreted in the same context, and they must be considered as a whole in order to optimize performance and prevent injuries. Processing systems will produce an alteration in the stimulus that generates physiological responses, so the manipulation of one of these three constructs will provoke changes in the others. With this objective, future studies that delve into this paradigm will be justified.

### Funding

This study was funded by the Ministry of Science and Innovation (Ministerio de Ciencia e Innovación) of the Spanish government (Grant Number: PID2019-107473RB-C21). The funders had no role in study design, data collection and analysis, decision to publish, or preparation of the manuscript.

### Grant Disclosures

The following grant information was disclosed by the authors:
Ministry of Science and Innovation (Ministerio de Ciencia e Innovación): PID2019-107473RB-C21.

## Competing Interests

The authors declare that they have no competing interests.

## Author Contributions

- Joan Fuster conceived and designed the experiments, performed the experiments, analyzed the data, prepared figures and/or tables, authored or reviewed drafts of the paper, and approved the final draft.
- Toni Caparrós conceived and designed the experiments, performed the experiments, authored or reviewed drafts of the paper, and approved the final draft.
- Lluis Capdevila conceived and designed the experiments, authored or reviewed drafts of the paper, and approved the final draft.

## Data Availability

No code was used in this Literature Review.

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
