# Peer review of "Evaluation of cognitive load in team sports: literature review"

_PeerJ, doi:10.7717/peerj.12045_

## Round 0.1 · original submission · Major Revisions

I agree with the reviewers that your research can provide a valuable contribution. I also agree with reviewers that the manuscript needs substantial revising prior to publication. Please attend closely to the reviewers' comments as you revise.

Reviewer 1 ·

Basic reporting

No comment

Experimental design

The review was well done, although it is necessary to clarify one point. Please watch my suggestions in general comments.

Validity of the findings

This point is developed in the general comments.

Additional comments

The keys to achieving the best sports training programme are complex because there are a lot of different variables involved. One of them is cognition which, again, involved other variables, but lately is gaining attention in both researchers and coaches because it has been observed that have an important influence on the performance of the athletes. In this regard, the authors did a literature review exploring how these cognitive variables were assessed, and what results were observed. I would like to congratulate them for the good work you have done in the manuscript. However, there are several aspects that need to improve to consider for publication. Below there is a list of suggestions that I believe would improve the paper's quality.
• Introduction: One of the points that this review could be of interest to the experts of the field is that the results of this research line are rather incongruent, and that is mainly because of the difficulty to control the variables. This incongruence produces confusion between those experts, and reviews like this should help to clarify the possible effects. I believe that this information should be added in a new paragraph
• Criteria exclusion: I noted that the authors excluded laboratory studies which, indeed, I believe is a good decision because it is far from real environments. However, these laboratory studies have been taken into account in this research line because of their capacity to control the variables. Certainly, one of the problems in the practical researches is that some results are incongruent, and one of the possible reasons is the lack of control of some variables. Did the authors consider the level of control of the variables? If so, please include it in the next version of the manuscript.
• I do not understand line 151. The following scale was only applied after the session? Then, why NASA was not included here? Or did NASA was applied before and after the session? Same case in line 175… My apologies. After reading it carefully, now I understand that it is the final line of that variable in which the authors specify the period in which the variable was assessed. Perhaps it is better to write that information at the end of the paragraph, or at the beginning, and not in italics?
• Line 355, please combine the brackets.

·

Basic reporting

General Comments:

The reviewer commends the authors on addressing such an important topic. As is stated in the introduction, cognitive load is known to influence physical performance, yet its existence is rarely, if ever, acknowledged when considering training load in sport. Further, I do not know of any literature which aims to address this issue.

That being said, there are several issues which detract from this important work in its current state. One general issue is that the outcomes of your literature search i.e., tools that aim to assess cognitive load and 'indicators' of cognitive load are not discussed until the very end of the introduction and not elaborated on. As this forms the major part of your manuscript I suggest increasing the focus on this. Please see my further specific comments below.

Specific Comments:

Lines 36-37 - It is unclear why change of direction and cognitive requirements are coupled together? Is change of direction not just as much a physical task as the jumps or accelerations mentioned above?

Line 37 – I suggest you either provide a reference for the statement ‘cognitive requirements and changes of direction are also identified….’ Or remove the words ‘identified as’

Lines 43-55 – I enjoyed this section. This paragraph is well written and speaks to the importance of this topic.

Line 53 – Define is not the correct term here. Something like ‘conclude’ would be more suitable.

Lines 56-57 – Is this how you are defining cognitive load for this review? If so, I suggest making this more explicit in this sentence.

Lines 60-62 – I agree with the authors on this point; however, it is unclear how this relates to cognitive load. This concept could be further explained.

Line 68-70 – This sentence is not clear. Please reword.

Line 81 – The term/concept of readiness is only introduced here. I suggest introducing earlier and explaining what you mean by this term.

Lines 86-89 – Reading to this point I was unsure what the aims of the review would be. Can these aims be introduced earlier or the earlier sections lead to the aims more explicitly? Further, it is not clear what is meant by an ‘indicator’. An explanation of this, and/or an example would increase clarity.

Experimental design

The methodology section requires more detail, as well as context or rationale, for a number of methodological decisions i.e.,

Line 93 - Why was the search limited to literature from 1970 onward?

Line 106 - Can you explain in a little more detail what is meant by 'management of cognitive, mental or psychological load'.

Line 112 - What is considered a 'real sport'

What was the screening process i.e., did one author screen all articles? Completed as a team?

Validity of the findings

The findings, although interesting and valid, need more context within the results/discussion. The broad array of studies included, i.e., different sports, different aims of studies etc make it difficult to follow and combine or compare the findings of the different studies. A greater description of the studies referenced is needed in text, not just in tables.

Further, the discussion largely describes the tools used to assess cognitive load. This is not particularly beneficial to the reader, rather a discussion of the pro's and con's of such measure might be more useful.

The discussion would also benefit from thorough proof reading and editing.

Lines 121-123 - As mentioned previously, an explanation of what is meant by an 'indicator' is required. As well as your classifications of indicators.

Lines 126-127 - This sentence needs more explanation and context.

Line 129 - I don't understand what is meant by 'analysis of indicators of response'.

Line 144 - Is this study a laboratory study? Laboratory studies were part of the exclusion criteria. There appears to be a number of lab studies included i.e., Smith (2016)

Line 153 - No need for the word scale. The S in VAS is scale.

Line 161 - Is bibliography the correct word here?

Lines 166-173 - This section does not really discuss the VAS, which is the subheading this section falls under.

Line 177 - Does CSAI stand for anything?

Line 184-188 - This section needs more explanation. E.g., what is meant by somatic and cognitive anxiety.

Line 192 - Spell out the full name for POMS at first use.

Line 203 - Change "In a study...". In what study?

Line 237 - Reference for this statement?

Line 244 - This full naming of SIATE needs to occur when this acronym is first used.

Line 285 - We or other authors?

Lines 294-295 - Physical fatigue could also increase reaction time.

Lines 297-301 - Here and elsewhere, these studies need describing. This information holds no value without greater explanation or context.

The conclusion brings in a lot of new information. A conclusion typically summarises the information stated previously. I would also suggest having the practical applications section, prior to the conclusion.

---

## Round 0.2 · accepted · Accept

Thank you for doing a thorough job addressing the reviewers' comments.

Reviewer 1 ·

Basic reporting

In general notes, the quality of this new version of the manuscript has been greatly increased. Therefore, I congratulate the authors for their implication in performing the necessary changes. I appreciate that my suggestions have been taken into account.
Congratulations of your excellent work.

Experimental design

No commments

Validity of the findings

No commments

Additional comments

No commments